# Prognostic role of perioperative acid-base disturbances on the risk of *Clostridioides difficile* infection in patients undergoing cardiac surgery

**Anna Rzucidło-Hymczak**[1], **Hubert Hymczak**[2], **Anna Kędziora**[3], **Bogusław Kapelak**[3], **Rafał Drwiła**[4], **Dariusz Plicner**[5]*

**1** Department of Pediatric Infectious Diseases and Pediatric Hepatology, John Paul II Hospital, Krakow, Poland, **2** Department of Anesthesiology and Intensive Care, John Paul II Hospital, Krakow, Poland, **3** Department of Cardiovascular Surgery and Transplantation, John Paul II Hospital, Krakow, Poland, **4** Jagiellonian University Medical College, Krakow, Poland, **5** Unit of Experimental Cardiology and Cardiac Surgery, Faculty of Medicine and Health Sciences, Andrzej Frycz Modrzewski Krakow University, Krakow, Poland

* plicner.dariusz@gmail.com

## Abstract

### Background

It is unclear whether acid-base balance disturbances during the perioperative period may impact *Clostridium difficile* infection (CDI), which is the third most common major infection following cardiac surgery. We hypothesized that perioperative acid-base abnormalities including lactate disturbances may predict the probability of incidence of CDI in patients after cardiac procedures.

### Methods

Of the 12,235 analyzed patients following cardiac surgery, 143 (1.2%) developed CDI. The control group included 200 consecutive patients without diarrhea, who underwent cardiac procedure within the same period of observation. Pre-, intra and post-operative levels of blood gases, as well as lactate and glucose concentrations were determined. Postoperatively, arterial blood was drawn four times: immediately after surgery and successively; 4, 8 and 12 h following the procedure.

### Results

Baseline pH was lower and $PaO_2$ was higher in CDI patients ($p < 0.001$ and $p = 0.001$, respectively). Additionally, these patients had greater base deficiency at each of the analyzed time points ($p < 0.001$, $p = 0.004$, $p = 0.012$, $p = 0.001$, $p = 0.016$ and $p = 0.001$, respectively). Severe hyperlactatemia was also more common in CDI patients; during the cardiac procedure, 4 h and 12 h after surgery ($p = 0.027$, $p = 0.004$ and $p = 0.001$, respectively). Multivariate logistic regression analysis revealed that independent risk factors for CDI following cardiac surgery were as follows: intraoperative severe hyperlactatemia (OR

**Data Availability Statement:** All relevant data are within the manuscript and its Supporting information files.

**Funding:** This article was supported by science found on John Paul II Hospital, Krakow, Poland (no. FN1/2021 to D.P.).

**Competing interests:** The authors have declared that no competing interests exist.

2.387, 95% CI 1.155–4.933, $p = 0.019$), decreased lactate clearance between values immediately and 12 h after procedure (OR 0.996, 95% CI 0.994–0.999, $p = 0.013$), increased age (OR 1.045, 95% CI 1.020–1.070, $p < 0.001$), emergent surgery (OR 2.755, 95% CI 1.565–4.848, $p < 0.001$) and use of antibiotics other than periprocedural prophylaxis (OR 2.778, 95% CI 1.690–4.565, $p < 0.001$).

## Conclusion

This study is the first to show that perioperative hyperlactatemia and decreased lactate clearance may be predictors for occurrence of CDI after cardiac surgery.

## Introduction

*Clostridioides difficile* (CD) is an anaerobic, Gram-positive bacillus, which may be part of the normal intestinal microbiota in healthy people. However, approximately 15% of adults experience colonization by CD and the prevalence is several times higher in hospitalized patients and in long-term care facilities residents [1]. CD is the most common cause of hospital-acquired diarrhea leading to increased morbidity and mortality in surgical patients [2]. In the last decades, the incidence of *Clostridium difficile* infection (CDI) has increased markedly worldwide [3, 4]. CDI is the third most common major infection (after pneumonia and bloodstream infections) following cardiac surgery [5].

There are many well-established risk factors for CDI development. These may include host factors (immune status, comorbidities), exposure to CD spores (hospitalizations) and other factors that disrupt normal colonic microbiome (antibiotics and other medications or surgery) [6]. It has also been shown that high glucose levels and stress hyperglycemia during the early postoperative period were associated with greater risk for development of CDI in patients following cardiac surgery [7]. It is unclear whether other acid-base disturbances, especially the development of hyperlactatemia in the perioperative period, may impact CDI occurrence.

Hyperlactatemia is a common occurrence in cardiac surgery and affects about 10 to 20% of patients [8]. Lactate is a product of pyruvate reduction by the enzyme lactate dehydrogenase during glycolysis. It is produced during physiological processes and is cleared by the liver and the kidney. However, in critically conditions associated with tissue hypoxia and anaerobic metabolism, pyruvate is accumulated rapidly and its metabolism is shifted to lactate production [9]. Hyperlactatemia can also result from reduced clearance, thus when increased production of lactate coexists with decreased clearance, the severity of the hyperlactatemia escalates [10]. An elevated lactate level can have profound hemodynamic consequences and is a well-recognized marker of circulatory failure and tissue hypoxia [11, 12] as well as being a sensitive and specific indicator of intestinal ischemia [13, 14]. Early onset of hyperlactatemia which develops intraoperatively or within the first 6 hours after surgery is associated with an increased risk for worse outcomes, prolonged hospital stay and death [8, 11, 12].

To the best of our knowledge, there have been no studies investigating the impact of acid-base balance disturbances on CDI occurrence following cardiac surgery. In the present report we tested the hypothesis that perioperative acid-base abnormalities, including lactate disturbances, may predict the probability of incidence of CDI in patients after cardiac procedures.

## Materials and methods

### Patients

This retrospective study was conducted by reviewing the medical records of 12,235 adult patients who underwent cardiac surgery in our institution from January 2014 to December 2019. The final study population comprised 143 patients who developed CDI during the post-operative period. The control group included 200 consecutive patients without diarrhea, who underwent cardiac surgery within the same period of observation. CDI was suspected in each patient experiencing three or more unformed stools per day and it was defined as a combination of symptoms and signs of the disease and confirmed by microbiological evidence of toxin-producing CD in the patients' stools [15]. Stool samples were analyzed using the rapid enzyme immunoassays test C, Diff Quik Chek Complete test (Techlab, Orlando, USA). Additionally, demographics, comorbidities, type and timing of cardiac surgery, perioperative infections and antibiotic treatment, readmission to intensive care unit and lastly in-hospital length of stay were collected.

Each patient received periprocedural antimicrobial prophylaxis, based on the first generation of cephalosporin (cefazolin). This was continued for another 3 to 5 doses postoperatively. Only in cases of history of allergy to cephalosporins or penicillin, clindamycin was administered.

The study was performed in accordance with the Declaration of Helsinki. The study was approved by the local Research Ethics Committee of the Andrzej Frycz Modrzewski Krakow University (ID 10/2019), which waived the need for inform consent due to the retrospective manner of analysis. Data were collected from Electronic Medical Records of John Paul II Hospital in Krakow (Poland), between January and March 2020. Personal identifiable information of the participants was anonymized upon extraction of the relevant data for the study, and patients were coded using numbers (1, 2, or 3, and so on).

### Laboratory investigations

Acid-base balance analyses were obtained from arterial lines that were placed in all patients before the cardiac procedure. Blood gases (pH, $PaCO_2$, $PaO_2$, base excess (BE)) as well as lactate and glucose concentrations were determined. Preoperative, intraoperative and postoperative levels of these parameters were assessed. Postoperatively, arterial blood was drawn four times: immediately after surgery and successively 4, 8 and 12 h following the procedure.

In our laboratory the reference ranges for normal values were as follows: for pH 7.35–7.45, for $PaCO_2$ 35–45 mmHg, for $PaO_2$ 74–108 mmHg, for BE -2.5–+2.5 mEq/l, for serum lactate concentration 0.5–1.6 mmol/l and for glucose concentration 3.9–5.5 mmol/l. The ranges were based on internal laboratory standardization for acid-base balance analyses and measurements were performed with ABL 835 FLEX blood gas analyzer (Radiometer Medical ApS, Brønshøj, Denmark).

Hyperlactatemia was defined as a peak lactate value > 2 mmol/l. Severe hyperlactatemia was diagnosed when peak lactate value was > 4 mmol/l based on our institution's laboratory reference ranges and a review of literature [8, 12, 16, 17]. Lactate clearance was calculated as follows: [(lactate initial– lactate delayed) /lactate initial] x 100% [18]. In this study the lactate clearances were calculated for the following intervals: between measurements preformed immediately and 4 h after surgery, between 4 h and 8 h following procedure, between 8 h and 12 h post operation, between values immediately and 12 h following surgery and finally between intraoperative values and 12 h after procedure.

Stress hyperglycemia was defined as one or more blood sugar concentration > 180 mg/dl (10 mmol/l) during the first 24 h of the postoperative course [5].

## Statistical analysis

Statistical analysis was performed using the IBM® SPSS® Statistics 25. Normal distribution was tested using the Kolmogorov-Smirnov and Shapiro-Wilk tests. Continuous variables were presented as mean and standard deviations (±) or median and interquartile ranges when appropriate. For categorical variables, numbers and proportions were reported. When appropriate, parametric and non-parametric tests were used for either independent samples (chi-squared test, Mann-Whitney U test, t-test) or repeated measurements (McNemar's test, Wicoxon signed-rank test, Firedman test). For ordinal variables, two sample Kolmogorov-Smirnov tests were used. Multivariate logistic regression model was calculated to determine independent predictors for CDI. A 2-tailed $p$ value of $< 0.05$ was considered to indicate statistical significance.

## Results

Of the 12,235 patients, 143 (1.2%) developed CDI. The CDI and control groups of analyzed patients were comparable, however patients with CDI were older in comparison to the control group (median age 71 vs 67, $p < 0.001$). Additionally, the CDI patients more often had a history of malignant neoplasms ($p = 0.048$) (Table 1).

## Acid-base balance

As shown in Table 2, patients with CDI had lower values of pH during the whole observation period, however, a significant difference was observed only during the preoperative period ($p < 0.001$, compared with the control group). There was no difference between groups in $PaCO_2$ levels in any of the studied periods, while $PaO_2$ was higher in CDI patients only at baseline ($p = 0.001$). Furthermore, at each of the analyzed time points, patients with CDI had greater base deficiency (more negative BE) ($p < 0.001$, $p = 0.004$, $p = 0.012$, $p = 0.001$, $p = 0.016$ and $p = 0.001$, respectively compared with the control group).

**Table 1. Baseline characteristics.**

| Variable | Patients with CDI (n = 143) | Patients without CDI (n = 200) | *p*-value |
|---|---|---|---|
| Age, (years) | 71 [64–77] | 67 [61–72] | **<0.001** |
| Male sex, n (%) | 93 (65) | 135 (67.5) | 0.634 |
| Comorbidities, n (%) | | | |
| Hypertension | 112 (78.3) | 160 (80) | 0.705 |
| Dyslipidemia | 69 (48.3) | 92 (46) | 0.680 |
| Diabetes mellitus | 44 (30.8) | 46 (23) | 0.107 |
| Chronic kidney disease | 32 (22.4) | 29 (14.5) | 0.060 |
| Atherosclerosis | 23 (16.1) | 19 (9.5) | 0.067 |
| Obesity | 21 (14.7) | 20 (10) | 0.187 |
| History of neoplasm | 15 (10.5) | 10 (5.0) | **0.048** |
| Peptic ulcer disease | 9 (6.3) | 12 (6.0) | 0.911 |
| Nicotinism | 5 (3.5) | 8 (4.0) | 0.810 |

Values are displayed as median [with 25th–75th percentiles inter-quartile range] or number (percentage). CDI: *Clostridium difficile* infection.

**Table 2. Variables of acid-base balance in both analyzed groups.**

| Variable | Patients with CDI (n = 143) | Patients without CDI (n = 200) | *p*-value |
|---|---|---|---|
| **pH** | | | |
| PREOP | 7.422 [7.372–7.445] | 7.434 [7.413–7.451] | **<0.001** |
| INTRA | 7.326 [7.269–7.376] | 7.336 [7.297–7.380] | 0.062 |
| H0 | 7.317 [7.273–7.369] | 7.324 [7.278–7.362] | 0.347 |
| H4 | 7.334 [7.266–7.380] | 7.337 [7.306–7.371] | 0.170 |
| H8 | 7.342 [7.278–7.389] | 7.354 [7.316–7.379] | 0.167 |
| H12 | 7.343 [7.286–7.375] | 7.348 [7.323–7.381] | 0.080 |
| **PaCO2 (mmHg)** | | | |
| PREOP | 36.80 [33.80–40.10] | 36.80 [34.60–38.77] | 0.721 |
| INTRA | 40.90 [36.90–44.60] | 41.35 [37.40–44.90] | 0.432 |
| H0 | 40.30 [36.90–44.60] | 40.80 [37.95–44.70] | 0.129 |
| H4 | 40.10 [35.80–43.70] | 40.70 [37.60–44.10] | 0.149 |
| H8 | 38.90 [36.10–42.10] | 39.10 [36.30–42.60] | 0.486 |
| H12 | 39.10 [36.17–43.90] | 39.80 [37.10–43.70] | 0.211 |
| **PaO2 (mmHg)** | | | |
| PREOP | 103.0 [82.1–221.0] | 87.7 [79.92–100.72] | **0.001** |
| INTRA | 204.0 [129.0–296.0] | 212.0 [136.50–302.75] | 0.900 |
| H0 | 165.0 [140.0–192.0] | 174.5 [133.25–202.75] | 0.252 |
| H4 | 163.0 [130.0–182.0] | 157.5 [122.50–182.0] | 0.436 |
| H8 | 151.0 [128.0–168.0] | 152.0 [125.0–174.0] | 0.735 |
| H12 | 144.0 [116.0–172.0] | 151.0 [127.0–171.0] | 0.155 |
| **BE (mEq/l)** | | | |
| PREOP | -0.60 [-2.90–1.50] | 0.7 [-0.8–1.8] | **<0.001** |
| INTRA | -5.10 [-7.30– -3.10] | -4.0 [-5.8– -1.7] | **0.004** |
| H0 | -4.90 [-7.10– -2.70] | -4.2 [-6.17– -2.1] | **0.012** |
| H4 | -4.80 [-6.60– -2.30] | -3.6 [-5.1– -1.8] | **0.001** |
| H8 | -4.20 [-6.50– -2.40] | -3.4 [-5.2– -1.5] | **0.016** |
| H12 | -3.95 [-6.95– -2.20] | -3.0 [-4.8– -1.5] | **0.001** |
| **Lactate (mmol/l)** | | | |
| PREOP | 1.0 [0.7–1.5] | 1.1 [0.9–1.47] | 0.126 |
| INTRA | 2.3 [1.7–3.7] | 2.2 [1.6–2.9] | 0.202 |
| H0 | 2.1 [1.3–3.6] | 2.0 [1.32–3.0] | 0.217 |
| H4 | 2.2 [1.3–4.3] | 1.9 [1.2–3.0] | **0.008** |
| H8 | 2.2 [1.4–4.1] | 1.8 [1.2–3.1] | **0.014** |
| H12 | 2.1 [1.2–4.6] | 1.6 [1.1–2.6] | **0.001** |
| **Lactate > 4 mmol/l, n (%)** | | | |
| PREOP | 5 (1.5) | 4 (1.2) | 0.393 |
| INTRA | 27 (7.9) | 21 (6.1) | **0.027** |
| H0 | 27 (7.9) | 23 (6.7) | 0.056 |
| H4 | 38 (11.1) | 28 (8.2) | **0.004** |
| H8 | 36 (10.5) | 37 (10.8) | 0.143 |
| H12 | 40 (12.0) | 25 (7.5) | **0.001** |
| **Lactate clearance (%)** | | | |
| 0–4 h | -1.72 [-42.86–15.00] | 6.46 [-18.04–23.08] | **0.003** |
| 4–8 h | -1.47 [-35.48–13.64] | 0.0 [-29.85–17.29] | 0.500 |
| 8–12 h | 9.09 [-9.09–23.19] | 10.666 [-13.33–31.58] | 0.130 |
| 0–12 h | 0.00 [-69.23–35.00] | 15.045 [-23.41–38.46] | **0.006** |

(*Continued*)

**Table 2.** (Continued)

| Variable | Patients with CDI (n = 143) | Patients without CDI (n = 200) | *p*-value |
|---|---|---|---|
| INTRA-12 h | 6.47 [-64.77–45.75] | 25.00 [-15.38–45.45] | **0.028** |
| **Glucose (mmol/l)** | | | |
| PREOP | 6.1 [5.4–7.7] | 6.0 [5.4–7.17] | 0.600 |
| INTRA | 9.4 [7.5–11.5] | 8.55 [7.2–10.4] | **0.032** |
| H0 | 9.0 [7.0–10.9] | 8.25 [6.9–10.2] | **0.040** |
| H4 | 8.7 [7.1–10.5] | 7.95 [6.7–9.4] | **0.019** |
| H8 | 8.6 [7.2–9.9] | 8.3 [7.2–9.6] | 0.498 |
| H12 | 8.3 [7.0–10.2] | 8.6 [7.2–9.6] | 0.758 |
| **Glucose >10 mmol/l, n (%)** | | | |
| PREOP | 18 (5.25) | 17 (4.96) | 0.218 |
| INTRA | 58 (16.90) | 62 (18.07) | 0.067 |
| H0 | 49 (14.28) | 52 (15.16) | 0.098 |
| H4 | 46 (13.41) | 41 (11.95) | **0.014** |
| H8 | 32 (9.36) | 38 (11.11) | 0.458 |
| H12 | 38 (11.41) | 37 (11.11) | 0.110 |

Values are displayed as median [with 25th–75th percentiles inter-quartile range] or number (percentage). CDI: *Clostridium difficile* infection; H0: measurements taken immediately after surgery; H4: measurements taken 4 h after surgery; H8: measurements taken 8 h after surgery; H12: measurements taken 12 h after surgery; INTRA: measurements taken during operation; PREOP: measurements taken before surgery.

Patients from the control group had the highest lactate concentration during surgery, then their lactate levels gradually decreased, whereas lactate clearance progressively increased. In CDI patients, lactate was also at maximum concentration during the procedure and remained elevated until the last observation. Additionally, patients with CDI at each of the analyzed time points, excluding the preoperative period, had higher lactate levels than the control group. During the postoperative course, in the last three measuring time points, this difference was significant ($p = 0.008$, $p = 0.014$ and $p = 0.001$, respectively). Severe hyperlactatemia was more common in CDI patients during the cardiac procedure, 4 h and 12 h after surgery ($p = 0.027$, $p = 0.004$ and $p = 0.001$, respectively, compared with the control group). The lactate clearance assessments were lower in patients with CDI during three intervals: between values immediately and 4 h after surgery, between measurements immediately and 12 h following surgery and between intraoperative values and 12 h after the procedure ($p = 0.003$, $p = 0.006$ and $p = 0.028$, respectively, compared with the control group). Patients with CDI also had higher glucose concentration than patients without CDI during the cardiac procedure, immediately and 4 h after surgery ($p = 0.032$, $p = 0.040$ and $p = 0.019$, respectively). Moreover, 4 h following surgery, patients with CDI more often had stress hyperglycemia ($p = 0.014$) (Table 2).

## Perioperative characteristics

There was no difference between analyzed groups in the type of operations (p = 0.448). The most common surgical procedures were heart valve surgery and coronary artery bypass grafting (for CDI patients: 37.8% and 37.1% and for non-CDI patients: 28.0% and 46.5%, respectively). There was also no difference in the type of perioperative antimicrobial prophylaxis used in the patients with and without CDI ($p = 0.537$) (Table 3).

Patients with CDI more frequently underwent emergent surgery, had longer cardiopulmonary bypass time and were more often readmitted to intensive care unit ($p < 0.001$, p = 0.010

**Table 3. Perioperative data.**

| Variable | Patients with CDI (n = 143) | Patients without CDI (n = 200) | *p*-value |
|---|---|---|---|
| Type of surgery, n (%) | | | 0.448 |
| HVS | 54 (37.8) | 56 (28) | |
| CABG | 53 (37.1) | 93 (46.5) | |
| Aortic surgery | 15 (10.5) | 9 (4.5) | |
| CABG + VHS | 10 (7.0) | 17 (8.5) | |
| CABG + aortic surgery | 6 (4.2) | 18 (9.0) | |
| MIDCAB | 5 (3.5) | 7 (3.5) | |
| Timing of surgery, n (%) | | | **<0.001** |
| Elective | 94 (65.7) | 167 (83.5) | |
| Emergent | 49 (34.3) | 33 (16.5) | |
| Cardiopulmonary bypass time, (min) | 106.0 [75.7–158.7] | 95.0 [72.0–120.0] | **0.010** |
| HVS | 123.0 [99.5–163.5] | 106.0 [91.0–129.0] | |
| CABG | 74.0 [60.0–93.0] | 78.0 [64.0–97.0] | |
| Aortic surgery | 200.0[120.0–225.0] | 137.0 [105.0–211.5] | |
| CABG + VHS | 135.0 [115.0–215.0] | 126.5 [97.5–147.2] | |
| CABG + aortic surgery | 205.0 [99.5–248.5] | 102.0 [78.2–126.0] | |
| Aortic cross-clamp time, (min) | 66.5 [37.2–91.5] | 58.0 [39.2–79.0] | 0.144 |
| HVS | 76.0 [63.0–98.5] | 71.0 [58.0–89.0] | |
| CABG | 37.0 [31.0–44.0] | 42.0 [33.5–54.5] | |
| Aortic surgery | 113.5 [83.0–162.0] | 89.0 [75.0–111.5] | |
| CABG + VHS | 87.0 [72.5–142.0] | 82.0 [69.2–99.2] | |
| CABG + aortic surgery | 88.0 [50.5–167.0] | 64.5 [29.0–80.0] | |
| Readmission to Intensive Care Unit, n (%) | 29 (20.3) | 18 (9.0) | **0.003** |
| Length of hospital stay, (days) | 22 [14.00–41.00] | 7 [6.00–9.75] | **<0.001** |
| Periprocedural prophylaxis based on Cefazolin, n (%) | 132 (92) | 188 (94) | 0.537 |
| Antibiotic other than periprocedural prophylaxis, n (%) | 71 (49.6) | 51 (25.5) | **<0.001** |
| Ceftriaxone | 45 (31.5) | 23 (11.5) | **<0.001** |
| Vancomycin | 20 (14) | 21 (10.5) | 0.326 |
| Fluoroquinolone | 18 (12.6) | 15 (7.5) | 0.115 |
| Piperacillin/Tazobactam | 13 (9.1) | 6 (3.0) | **0.015** |
| Meropenem | 10 (7.0) | 6 (3.0) | 0.084 |
| Ampicillin | 5 (3.5) | 6 (3.0) | 0.797 |
| Cloxacillin | 6 (4.2) | 3 (1.5) | 0.124 |
| Clindamycin | 5 (3.5) | 7 (3.5) | 0.999 |
| Gentamicin | 5 (3.5) | 4 (2.0) | 0.393 |
| Rifampicin | 5 (3.5) | 2 (1.0) | 0.107 |
| Amoxicillin/Clavulanic acid | 3 (2.1) | 2 (1.0) | 0.403 |
| Teicoplanin | 3 (2.1) | 1 (0.5) | 0.174 |
| Cefuroxime | 2 (1.4) | 0 (0) | 0.093 |
| Colistin | 2 (1.4) | 1 (0.5) | 0.378 |
| Erythromycin | 1 (0.7) | 0 (0) | 0.236 |
| Accompanying infection, n (%) | 67 (46.9) | 46 (23) | **<0.001** |
| Wound infection | 19 (13.3) | 13 (6.5) | **0.033** |
| Pneumonia | 16 (11.2) | 10 (5.0) | **0.033** |
| Sepsis | 39 (27.3) | 25 (12.5) | **0.001** |

(*Continued*)

**Table 3.** (Continued)

| Variable | Patients with CDI (n = 143) | Patients without CDI (n = 200) | *p*-value |
|---|---|---|---|
| Infective endocarditis | 5 (3.5) | 4 (2.0) | 0.393 |

Values are displayed as median [with 25th–75th percentiles inter-quartile range] or number (percentage). CDI: *Clostridium difficile* infection; HVS: heart valve surgery; CABG: coronary artery bypass grafting; MIDCAB: minimally invasive coronary artery bypass.

**Table 4. Multivariate logistic regression for risk of *Clostridioides difficile* infection.**

| Variable | OR (95% CI) | *p*-value |
|---|---|---|
| Age | 1.045 (1.020–1.070) | <**0.001** |
| Emergent surgery | 2.755 (1.565–4.848) | <**0.001** |
| Lactate clearance 0-12H | 0.996 (0.994–0.999) | **0.013** |
| INTRA Lactate > 4 mmol/l | 2.387 (1.155–4.933) | **0.019** |
| Antibiotic other than periprocedural prophylaxis | 2.778 (1.690–4.565) | <**0.001** |

OR: odds ratio; CI: confidence interval; INTRA: measurements taken during operation.

and $p = 0.003$, respectively, compared with the control group). Additionally, patients with CDI more often had accompanying infections ($p < 0.001$), such as wound infection, pneumonia and sepsis ($p = 0.033$, $p = 0.033$ and $p = 0.001$, respectively, compared with the control group). These patients also more often received additional antibiotics besides periprocedural antimicrobial prophylaxis ($p < 0.001$, compared with the control group). Ceftriaxone and piperacillin plus tazobactam were used more frequently in patients with CDI than in non-CD subjects ($p < 0.001$ and $p = 0.015$, respectively). The median length of hospital stay for CDI patients was 22 days [14.00–41.00] and 7 days [6.00–9.75] for patients without CDI ($p < 0.001$) (Table 3).

Multivariate logistic regression analysis revealed that independent risk factors for CDI following cardiac surgery were intraoperative severe hyperlactatemia (OR 2.387, 95% CI 1.155–4.933, $p = 0.019$), decreased lactate clearance between values immediately and 12 h after procedure (OR 0.996, 95% CI 0.994–0.999, $p = 0.013$), increased age (OR 1.045, 95% CI 1.020–1.070, $p < 0.001$), emergent surgery (OR 2.755, 95% CI 1.565–4.848, $p < 0.001$) and use of antibiotics other than periprocedural prophylaxis (OR 2.778, 95% CI 1.690–4.565, $p < 0.001$) (Table 4).

## Discussion

To the best of our knowledge, this study is the first to evaluate the association between perioperative changes in an acid-base balance and CDI incidence in patients after cardiac surgery. We demonstrated that perioperative increased lactate concentration and decreased lactate clearance may be independent predictors of occurrence of CDI in the analyzed group of patients.

### Acid-base balance

In this study, patients with CDI had higher lactate levels after surgery in comparison to the control group. The most significant difference was 12 h following surgery. Most cases of severe hyperlactatemia also occurred at that time. Moreover, patients with CDI had reduced lactate clearance, therefore hyperlactatemia was intensified by impaired lactate clearance.

There are many potential causes of hyperlactatemia in cardiac surgical patients [8]. One cause may be tissue hypoperfusion and anaerobic metabolism as a result of inadequate oxygen delivery during cardiopulmonary bypass. Other reasons of elevated lactate levels may be renal failure, shock, excessive administration of lactated Ringer's solution and use of catecholamines [8]. Hajjar et al. showed that high lactate levels at the end of the cardiac surgery and during the postoperative period can identify patients with worse postoperative outcomes including a higher rate of 30-day mortality [16]. Similarly, Maillet et al. proved that lactate threshold of 3 mmol/l at admission to the intensive care unit is able to identify a population at risk of morbidity and mortality after cardiac surgery [17]. Therefore, a targeting therapy to reduce or prevent the increase in lactate levels may improve outcomes of post cardiac surgery [8, 12, 16, 17]. It is also known that early onset hyperlactatemia that develops intraoperatively or within the first 6 hours after surgery is associated with an increased risk for worse hemodynamic outcomes, prolonged hospital stay and death, whereas late onset hyperlactatemia is associated with a benign postoperative course [8]. Our study showed that both early and late onset hyperlactatemia may affect the incidence of CDI.

Besides higher lactate levels, patients with CDI also had more negative BE throughout the perioperative period. An excessive negative BE result indicates an alkaline deficiency and best reflects metabolic acidosis after cardiac surgery. This may in part be explained by the fact that lactate is a strong anion that is virtually fully dissociated at physiological pH, and increased lactate concentration reduces the strong ion difference and exerts an acidifying effect on the blood [8]. It is also known that other underlying causes for acid-base disturbances after cardiac surgery are manifold and are best displayed by changes in BE [19].

We also demonstrated that patients with increased glucose levels during the operation and in the early postoperative period were at greater risk for development of CDI. This finding is consistent with the results of a study by Kirkwood et al., who demonstrated the association of acute hyperglycemia with an increased risk of CDI [20]. Similarly, Gelijns et al. showed that stress hyperglycemia was associated with major infection after cardiac surgery [5]. It should be emphasized that our study did not show that diabetes mellitus is associated with the risk of CDI. Our study only proved this association for abnormally high blood glucose levels during the perioperative course. Hyperglycemia induces an impairment of host defenses (e.g., damage to the neutrophil function, disturbances of the oxidant system and humoral immunity) and favor the greater frequency of infections [21]. Therefore, guidelines recommend a rigorous control of glycemia during the postoperative period to reduce surgical infections [22].

## Perioperative characteristics

There is significant evidence that many comorbidities increase the risk of CDI development. Several well-established risk factors such as older age, inflammatory bowel disease, gastrointestinal surgery and immunological incompetence caused by malignant neoplasms, transplantations, chronic kidney diseases and immunosuppressant therapy are associated with an increased occurrence of CDI [23–26]. Our results validate these findings and we also showed that older age and a history of malignant neoplasms were associated with increased risk of CDI.

Timing of surgery was also an important risk factor for the development of CDI. In our report the most cases of CDI occurred after emergent surgery. This finding is consistent with the results of Lemaire et al., who also demonstrated the significant role of emergent cardiac surgery in the development of CDI [27]. Our results also validate the findings of Gelijns et al., who suggested that longer cardiopulmonary bypass time had an impact on major infection after cardiac surgery including CDI [5].

In our study, besides periprocedural antimicrobial prophylaxis, some patients received an additional antibiotic due to accompanying infections other than CDI, and these patients had greater chance of contracting CDI. Infection of the wound, pneumonia and sepsis were the most likely factors correlated with CDI occurrence. In part, this finding could be explained by reduction in immune response to infections and the need for additional antibiotics in such patients. It is well known that the risk of CDI increases substantially with multiple, prolonged antibiotic exposure [27, 28]. The pathogenesis of CDI includes disruption of the host micro-biota, usually with broad-spectrum antibiotics, proliferation of toxins after germination of CD in the colon, and lack of immune response to the infection [29]. It should be highlighted that the majority of antibiotics may lead to development of CDI, yet most often it is caused by clin-damycin, third-generation cephalosporins, fluoroquinolones and broad-spectrum penicillins [28]. In our research, the analysis of the antibiotics used before the CDI also showed that ceftri-axone (third-generation cephalosporin) and piperacillin plus tazobactam (broad-spectrum penicillin) were the antibiotics mostly correlated with the development of postoperative CDI. A serious approach must be undertaken to reduce unnecessary and excessive administration of antibiotics in surgical patients thereby preventing the development of CDI.

The majority of reports confirm the finding presented in this study, that prolonged hospi-talization in the intensive care unit may be a significant risk factor of patients developing CDI [30, 31]. In our study patients who were readmitted to this ward had greater chance of infec-tion. Also, Šuljagić et al. showed that duration of intensive care unit stay could be a significant predictor of CDI in surgical patients [31]. A reason could be that this group of patients were in a worse general condition, had more comorbidities or co-infections and received additional antibacterial treatment. There are studies that indicate that CDI affects the length of hospital stay [20, 27]. We also proved that patients with CDI had longer median postoperative inpatient stay (from surgery to discharge). Longer hospitalization in CDI patients most likely increases the cost of hospitalization after cardiac surgery, but unfortunately we have not studied this.

## Limitations

This study has several limitations. It was a retrospective research, based on available medical data. Therefore, the incidence of CDI could be higher due to a lack of information regarding potential post-discharge diagnosis of the disease. The size of the study group was limited. Moreover, the relation between CDI and accompanying infections is not clear, because the time sequence of the development of other infections was not collected. We assumed that patients who developed an accompanying infection had been treated with antibiotics and therefore would be more susceptible to CDI. However, it is not known whether CDI or other infections developed first. At this time, it is clear that there is an association between CDI and wound infection, pneumonia and sepsis. A prospective study with postoperative follow-up would identify the time of development of CDI and accompanying infections and determine causality. Additionally, excess length of hospital stay due to CDI should be interpreted with caution, because we did not take into account other adverse events and complications (e.g. other infections, hemodynamic instability) after surgery which may have affected length of hospital stay. Finally, most importantly, in our study we did not investigate the causes of hyperlactatemia after cardiac surgery.

## Conclusion

In conclusion, our findings indicate that perturbations of the perioperative acid-base balance increase the risk of CDI in patients after cardiac surgery. Correlation between severe hyperlac-tatemia and impaired lactate clearance and CDI incidence might suggest that these markers

could be useful in identifying patients at higher risk of developing of CDI following cardiac procedures.

## Supporting information

**S1 File.**
(XLSX)

## Author Contributions

**Conceptualization:** Anna Rzucidło-Hymczak, Dariusz Plicner.

**Data curation:** Anna Rzucidło-Hymczak, Hubert Hymczak, Dariusz Plicner.

**Formal analysis:** Anna Rzucidło-Hymczak, Hubert Hymczak, Anna Kędziora, Dariusz Plicner.

**Investigation:** Anna Rzucidło-Hymczak, Hubert Hymczak, Dariusz Plicner.

**Methodology:** Anna Rzucidło-Hymczak, Dariusz Plicner.

**Project administration:** Anna Rzucidło-Hymczak.

**Resources:** Hubert Hymczak, Anna Kędziora, Bogusław Kapelak, Rafał Drwiła.

**Supervision:** Anna Rzucidło-Hymczak, Dariusz Plicner.

**Validation:** Anna Rzucidło-Hymczak, Dariusz Plicner.

**Writing – original draft:** Anna Rzucidło-Hymczak, Dariusz Plicner.

**Writing – review & editing:** Anna Rzucidło-Hymczak, Hubert Hymczak, Anna Kędziora, Bogusław Kapelak, Rafał Drwiła, Dariusz Plicner.

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
