## [Decision Letter · Decision Letter 0]

14 Jan 2021

PONE-D-20-38300

Prognostic role of perioperative acid-base disturbances on the risk of Clostridioides difficile infection in patients undergoing cardiac surgery

PLOS ONE

Dear Dr. Plicner,

Thank you for submitting your manuscript to PLOS ONE. After careful consideration, we feel that it has merit but does not fully meet PLOS ONE’s publication criteria as it currently stands. Therefore, we invite you to submit a revised version of the manuscript that addresses the points raised during the review process.

Please address the reviewers' issues and revise accordingly. 

We look forward to receiving your revised manuscript.

Kind regards,

Academic Editor

PLOS ONE

Journal Requirements:

2. In the ethics statement in the manuscript and in the online submission form, please provide additional information about the patient records/samples used in your retrospective study, includinga) the date range (month and year) during which patients' medical records/samples were accessed; b) the source of the medical records/samples analyzed in this work (e.g. hospital, institution or medical center name).

Reviewers' comments:

Reviewer's Responses to Questions

**Comments to the Author**

1. Is the manuscript technically sound, and do the data support the conclusions?

Reviewer #1: Yes

Reviewer #2: Yes

2. Has the statistical analysis been performed appropriately and rigorously? 

Reviewer #1: Yes

Reviewer #2: Yes

3. Have the authors made all data underlying the findings in their manuscript fully available?

Reviewer #1: Yes

Reviewer #2: Yes

4. Is the manuscript presented in an intelligible fashion and written in standard English?

Reviewer #1: Yes

Reviewer #2: Yes

5. Review Comments to the Author

Reviewer #1: I would like to commend the authors on this study. CDiff infections continue to be a morbidity in surgical fields and it is important for us to better understand it in cardiac surgery. There are a couple of issues I would like to communicate with this manuscript.

1. Please indicate what type of operations the patients undergo and the clamp and bypass times for these operations. It would be important to know if increased bypass and clamp times have an effect on cdiff operations.

2. Having different types of preoperative antibiotics may have an effect on cdiff infections. What is the reasoning for using these different types of antibiotics instead of having the standard. How about post-op antibiotics. At our institution, we usually continue antibiotics for another 3-5 doses post-operatively.

3. It is possible to take a look at patients who did not have postoperative infections requiring antibiotics. This way, we can isolate those patients who did not require escalation in antibiotics who developed cdiff. We already know additional or escalating antibiotics is a risk factor for cdiff infections. In the same manner, should we considering doing the same with endocarditis patients.

Thank you for the manuscript.

Reviewer #2: Dear the authors of the manuscript entitled "Prognostic role of perioperative acid-base disturbances on the risk of Clostridioides difficile infection in patients undergoing cardiac surgery"

Thank you for writing this manuscript which highlights an important issue in the field of cardiac surgery. it is unique experiment and well formed. the manuscript is well written and the results and conclusion are beneficial to the readers

I have no concerns

thank you

6. PLOS authors have the option to publish the peer review history of their article (what does this mean?). If published, this will include your full peer review and any attached files.

Reviewer #1: No

Reviewer #2: **Yes: **salah eldien Altarabsheh

---

## [Author Response · Author response to Decision Letter 0]

11 Feb 2021

Dear Reviewer of PLOS ONE,

Thank you very much for your patience and all the valuable comments and suggestions. I tried my best to address all your concerns. Please find my answers below.

Sincerely yours,

Dariusz Plicner on behalf of the authors.

1. Please indicate what type of operations the patients undergo and the clamp and bypass times for these operations. It would be important to know if increased bypass and clamp times have an effect on cdiff operations.

Ad. 1 We performed additional calculations according to your recommendations.

The most common surgical procedures performed in the analyzed groups of patients were heart valve surgery and CABG (for CDI patients: 37.8% and 37.1% and for non-CDI patients: 28.0% and 46.5%, respectively). There was no difference between groups in the type of operations (p = 0.448). 

Patients with CDI had longer cardiopulmonary bypass time than the control group (p = 0.01), but there were no correlations between CDI occurrence and aortic cross-clamp time (p = 0.144 ).

We added the above findings to the results section (p. 10, lines 176-178, 181-182 and Table 3) and discussion (p. 14, lines 258-260).

2. Having different types of preoperative antibiotics may have an effect on cdiff infections. What is the reasoning for using these different types of antibiotics instead of having the standard. How about post-op antibiotics. At our institution, we usually continue antibiotics for another 3-5 doses post-operatively.

Ad. 2 At our institution, each patient who undergoes cardiac surgery receives periprocedural antimicrobial prophylaxis, based on the first generation of cephalosporins (cefazolin). This is also continued for another 3-5 doses postoperatively. Only in cases of history of allergy to cephalosporins or penicillin, clindamycin was administered. There was no difference in the type of perioperative antimicrobial prophylaxis used in the patients with and without CDI (92% vs 94 %, respectively, p = 0.537). 

In compliance with your suggestions we added this information to the methods section (p. 5, lines 92-94) and results (p. 10, lines 179-180).

3. It is possible to take a look at patients who did not have postoperative infections requiring antibiotics. This way, we can isolate those patients who did not require escalation in antibiotics who developed cdiff. We already know additional or escalating antibiotics is a risk factor for cdiff infections. In the same manner, should we considering doing the same with endocarditis patients.

Ad. 3 Thank you for this valuable comment. It offers a very interesting and broad subject for our next study. We already have a large group of patients collected from 2014. We also believe that this is an important area to study which is why we mentioned this problem in the limitations section (p. 16, lines 292-298).

---

## [Decision Letter · Decision Letter 1]

1 Mar 2021

Prognostic role of perioperative acid-base disturbances on the risk of Clostridioides difficile infection in patients undergoing cardiac surgery

PONE-D-20-38300R1

Dear Dr. Plicner,

We’re pleased to inform you that your manuscript has been judged scientifically suitable for publication and will be formally accepted for publication once it meets all outstanding technical requirements.

Kind regards,

Academic Editor

PLOS ONE

Additional Editor Comments (optional):

Reviewers' comments:

Reviewer's Responses to Questions

**Comments to the Author**

1. If the authors have adequately addressed your comments raised in a previous round of review and you feel that this manuscript is now acceptable for publication, you may indicate that here to bypass the “Comments to the Author” section, enter your conflict of interest statement in the “Confidential to Editor” section, and submit your "Accept" recommendation.

Reviewer #1: All comments have been addressed

Reviewer #2: All comments have been addressed

2. Is the manuscript technically sound, and do the data support the conclusions?

Reviewer #1: Yes

Reviewer #2: Yes

3. Has the statistical analysis been performed appropriately and rigorously? 

Reviewer #1: Yes

Reviewer #2: Yes

4. Have the authors made all data underlying the findings in their manuscript fully available?

Reviewer #1: Yes

Reviewer #2: Yes

5. Is the manuscript presented in an intelligible fashion and written in standard English?

Reviewer #1: Yes

Reviewer #2: Yes

6. Review Comments to the Author

Reviewer #1: Thank you for revising your manuscript and addressing all of my points in a point by point fashion. Obviously, there are limitations to your work and you have addressed them in the limitations section. I believe your further studies will be very interesting. THank you for submitting your work.

Reviewer #2: Thank you very much for taking care of the reviewers comments. I have no concerns about this manuscript

7. PLOS authors have the option to publish the peer review history of their article (what does this mean?). If published, this will include your full peer review and any attached files.

Reviewer #1: No

Reviewer #2: **Yes: **salah eldien Altarabsheh

---

## [Editor Report · Acceptance letter]

8 Mar 2021

PONE-D-20-38300R1 

Prognostic role of perioperative acid-base disturbances on the risk of *Clostridioides difficile* infection in patients undergoing cardiac surgery 

Dear Dr. Plicner:

I'm pleased to inform you that your manuscript has been deemed suitable for publication in PLOS ONE. Congratulations! Your manuscript is now with our production department. 

Kind regards, 

on behalf of

Dr. Robert Jeenchen Chen 

Academic Editor

PLOS ONE